# Comprehensive Artistic Style Representation for Quantitative Evaluation

## Abstract

Artistic style, a unique medium for artists to express creativity through elements like form, color, and composition, poses a challenge for computer vision due to its intricate patterns and nuanced aesthetics. Contemporary models, often reliant on specific datasets, face limitations in their generalizability and precision in identifying individual artists' styles. From an information theory perspective, we examine the limitations of fine-tuning and investigate techniques to disentangle content from style information. We note differences in artistic style representation between unimodal and multimodal models. As a result, we propose a plug-and-play approach designed to efficiently separate content information within Vision-Language Models (VLMs), preserving stylistic details. Furthermore, we present the WeART dataset, a large-scale art dataset with high-quality annotations, to evaluate the artistic style representation capabilities of models. Experimental results show that our method improves the performance of VLMs in style retrieval tasks across several datasets. We will publicly release the proposed dataset and code.

## 1 Introduction

*"Style is a simple way of saying complicated things."* – Jean Cocteau

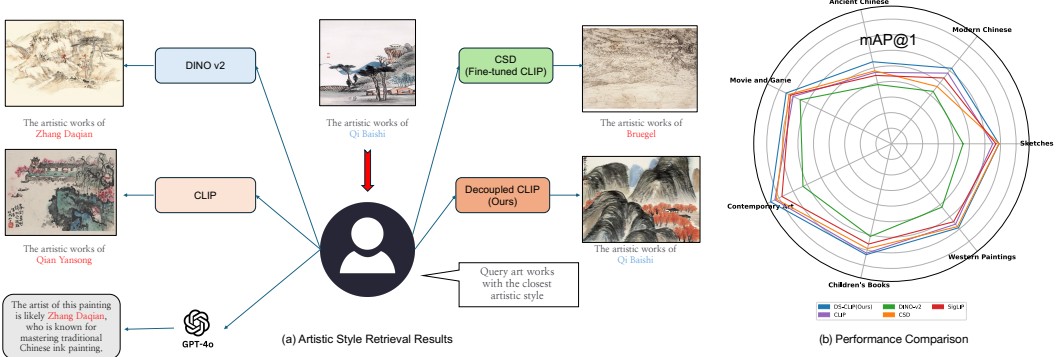

Figure 1: (a) Style representation models are typically divided into two categories: general representation models and specific style fine-tuning models. General representation models struggle to distinguish between content and artistic style when mixed, leading to retrieval results with similar content but different styles. Specific style fine-tuning models are limited by the diversity of training datasets, performing poorly with diverse artistic styles. In contrast, our model achieves more accurate and extensive style representation by decoupling multimodal and unimodal visual representations. (b) Performance comparison between our model and other retrieval baseline models.

Artistic style is the essence of an artist's personalized expression, conveyed through visual elements such as form, color, and composition to communicate themes and creativity. Despite advancements in computer vision technology, recognizing and describing this unique style remains challenging, as illustrated in Figure 1(a). Artistic style encompasses complex visual patterns, aesthetic preferences, and unique techniques, which are difficult to capture through traditional quantitative analysis

or natural language descriptions. With the development of generative models, accurately expressing artistic style in tasks such as image transformation, style transfer, and editing has become crucial. However, the subjectivity and personalized techniques inherent in artistic style make it challenging for general image representation methods to capture these subtle differences, often leading to confusion between style and content.

Currently, most artistic style representation models Wright & Ommer (2022); Wang et al. (2023); Somepalli et al. (2024); Zhang et al. (2019); Girdhar et al. (2023); Li et al. (2023) rely on classification datasets of specific art forms for training. The effectiveness of these methods is limited by the richness of the datasets, making it difficult to handle a broader range of art categories. Consequently, these models have limited generalization ability and practicality when faced with unknown artistic styles or categories. Moreover, current models often face challenges in fine-tuning their style representations down to the granularity of individual artists, which results in challenges when it comes to precisely discerning stylistic variations among works by different artists.

We identify noticeable differences in the representation of artistic styles between multimodal Vision-Language Models (VLMs) Radford et al. (2021); Li et al. (2022); Zhai et al. (2023); Zhu et al. (2023) and unimodal self-supervised modelsCaron et al. (2021); Oquab et al. (2023); Chen* et al. (2021). VLMs can simultaneously capture both semantic information and stylistic features of images, whereas unimodal self-supervised models excel at recognizing the content representation of images. Leveraging this disparity, we conduct an in-depth analysis of the overlapping areas between style and content in image information representation, as well as the overlapping areas between different modalities in multimodal models. Based on this analysis, we propose a visual representation decoupling technique grounded in mutual information. The core of this method is to retain the stylistic features of the image while removing non-stylistic content information, enabling the model to focus more effectively on representing artistic style.

To evaluate the capability of existing models to represent artistic style, we introduce a new high-quality dataset named WeART. This dataset contains 197,632 images from public museum collections, artists' public blogs, and personal portfolios, covering seven art categories and including works by 661 internationally renowned artists. Through extensive screening and strict selection criteria, WeART ensures the high-definition quality and integrity of each artwork. In terms of scale, diversity, and image quality, WeART provides valuable resources for the study of artistic style and promotes further research and technological advancements in this field.

Experimental results demonstrate that our method adheres to the principles of mutual information analysis and achieves performance enhancements in style retrieval tasks, as shown in Figure 2(b), across two annotated artist datasets, WikiART Saleh & Elgammal (2015) and WeART. Notably, our decoupling approach is designed to be modular and flexible, ensuring compatibility with a variety of VLMs and fine-tuned style representation models. This not only enhances their artistic representation capabilities but also improves the robustness and adaptability of these models.

In summary, the main contributions of this paper include:

- Analyze existing artistic style and general image representation models from an information theory perspective, identifying key criteria for effective style models.

- Develop a comprehensive decoupling method that isolates content-independent artistic styles from VLMs, enhancing the model's stylistic representation.

- Introduce the WeART dataset, which improves scale, diversity, and image quality over existing art datasets. Experiments show our method's effectiveness on both WikiART and WeART datasets, demonstrating cross-model compatibility.

## 2 RELATED WORK

### 2.1 STYLE REPRESENTATION LEARNING

Learning style representation Graham et al. (2012); Lun et al. (2015); Matthews & Merriam (2020); Silva et al. (2021); Srinivasa Desikan et al. (2022) is both challenging and crucial. Effective style representation enhances our understanding of artistic styles and supports applications such as fine-grained image retrieval, creative generation guidance, and copyright infringement detection. Early

methods relied on basic visual attributes, including color distribution, texture patterns, and compositional structure. Specific characteristics unique to different art forms, such as oil paintings or Chinese paintings, were employed to capture distinctive stylistic features.

For instance, ArtFID Wright & Ommer (2022) compiled multiple art datasets and trained a classification network, proposing a style evaluation metric. Similarly, Lee et al. Lee et al. (2021) utilized separate neural network modules for image style and content to facilitate style-based image retrieval. Wang et al. Wang et al. (2023) developed an attribution model trained on synthesized styles to identify images with similar styles. Additionally, Li et al. Li et al. (2023) employed the Gram matrix to extract textural features and clustered these into a style space. SomePalli et al. Somepalli et al. (2024) introduced a framework for extracting style descriptors. However, these efforts are based on fine-tuning datasets of specific artists, limiting their applicability to out-of-distribution (OOD) art styles.

## 2.2 STYLE TRANSFER

Artistic style transfer Dumoulin et al. (2016); Gatys et al. (2016); Huang & Belongie (2017); Luan et al. (2017); Park et al. (2020); Wang et al. (2022); Zhang et al. (2013) aims to apply the artistic style of one image to another, generating a new image with a specific style. Gatys et al. Gatys et al. (2016) introduced a method based on Convolutional Neural Networks (CNNs) in 2015, optimizing content and style features, which quickly gained attention.

With deep learning advancements, style transfer methods have evolved. Improved algorithms like Fast Style Transfer Johnson et al. (2016) and Arbitrary Style Transfer Huang & Belongie (2017) enhance computational efficiency and transfer effectiveness. Technologies like Generative Adversarial Networks (GANs) Goodfellow et al. (2014) and Variational Autoencoders (VAEs) Kingma (2013) have further improved the quality and diversity of generated images. Recently, Diffusion Models Ho et al. (2020) have shown significant potential, generating high-quality images by progressively adding noise and learning the denoising process.

However, the current landscape is marked by an absence of objective and quantifiable benchmarks for assessing style transfer efficacy. Traditional metrics for image quality assessment, such as the Fréchet Inception Distance (FID)Heusel et al. (2017) and the Inception Score (IS)Barratt & Sharma (2018), fail to directly measure the stylistic resemblance between the generated image and its reference counterpart.

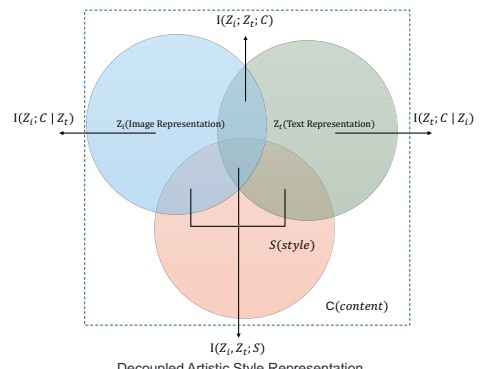

Figure 2: We utilize mutual information analysis to separate artistic style representations. Three circles represent image features from VLMs, text features, and style-related information; their intersections illustrate shared mutual information. By mitigating content-related feature, this method enables the precise extraction of style features.

## 3 MOTIVATION AND ANALYSIS

We start by providing a formal definition of the modalities and information within image-text models: we denote the image modality as $X_i$, the textual modality as $X_t$, style-related information as $S$, and style-unrelated information, which we consider as content information, as $C$. As shown in Figure 2, Using the definition of mutual information, we can partition the style-related information within image-text models into three components, as shown in equation (1):

$$I(X_i, X_t; S) = I(X_i; X_t; S) + I(X_i; S \mid X_t) + I(X_t; S \mid X_i). \tag{1}$$

$I(X_i; X_t; S)$ represents the shared style information, $I(X_i; S \mid X_t)$ represents the style information exclusive to the image modality. Similarly, $I(X_t; S \mid X_i)$ signifies the style information carried

solely by the text modality. At the same time, the style-related information $I(X_i, X_t; S)$ in the image can be obtained by decoupling and removing the content-related information from the total mutual information, as shown in Equation (2):

$$I(X_i, X_t; S) = I(X_i; X_t) - I(X_i, X_t \mid S) = I(X_i; X_t) - I(X_i, X_t; C). \quad (2)$$

Current contrastive learning based VLMs primarily focus on increasing the mutual information $I(X_i; X_t)$ between images and text, and further enhancing task-relevant information $I(X_i; X_t; S)$ during the supervised fine-tuning phase, yet they neglect the modeling of unique information in images and texts. Typically, such methods generate a pair of representations Tosh et al. (2021); Tsai et al. (2020) as:

$$Z_i = \underset{Z_i := E_{img}(X_i)}{\arg\max} \ I(Z_i; X_t), \quad Z_t = \underset{Z_t := E_{txt}(X_t)}{\arg\max} \ I(Z_t; X_i). \quad (3)$$

$Z_i$ can encode images $X_i$ and $Z_t$ can encode text $X_t$, by maximizing a lower bound on $I(X_i; X_t)$ using the InfoNCE loss. From the standpoint of information theory Liang et al. (2021; 2024), we are able to obtain:

$$I(Z_i, Z_t; S) = I(X_i, X_t; S) - I(X_i; S \mid Z_t) - I(X_t; S \mid Z_i) < I(X_i, X_t; S). \quad (4)$$

Comprehensive and extensive datasets can effectively reduce the information gap between $X$ and $Z$. However, fine-tuning with small-scale data often exaggerates this gap, thereby impacting the model's applicability and robustness. Decoupling style representations from the image representations of large-scale trained VLMs is a more general approach. We can decompose the mutual information that is relevant to content but irrelevant to style into the following three components:

$$\begin{aligned} I(Z_i, Z_t; S) &= I(Z_i; Z_t) - I(Z_i, Z_t; C) \\ &= I(Z_i; Z_t) - I(Z_i; Z_t; C) - I(Z_i; C \mid Z_t) - I(Z_t; C \mid Z_i). \end{aligned} \quad (5)$$

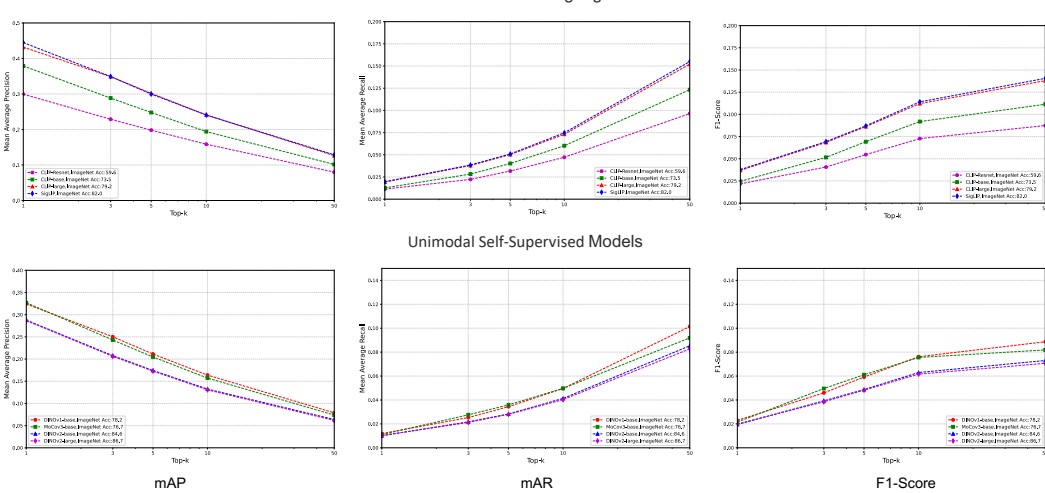

Figure 3: Multimodal VLMs exhibit strong performance on ImageNet also demonstrate excellence in artistic style retrieval on the WikiART dataset. In contrast, Unimodal self-supervised models show an opposing trend.

As illustrated in Figure 3, we observe that VLMs exhibit characteristics that are markedly different from those of self-supervised image representation models. Due to variations in training data, data augmentation strategies, and evaluation methods, VLMs that perform well on ImageNet also excel in artistic style retrieval tasks. In contrast, self-supervised models demonstrate the opposite trend.

Self-supervised image models are trained using only the image modality, and robust data augmentation strategies effectively disrupt the style information while preserving the content information. Consequently, we believe that self-supervised models, such as DINOv2, contain more content information exclusive to the image side compared to multimodal models like CLIP.

Furthermore, we note that in the text representations of multimodal models, the content features described by the text include not only the shared information between images and text but also information exclusive to the text. Therefore, our research aims to decouple and obtain representations related solely to style by removing the image-exclusive information $I\left(Z_i; C \mid Z_t\right)$ contained in DINOv2 from the multimodal models, as well as the shared information $I\left(Z_i; Z_t; C\right)$ and text-exclusive information $I\left(Z_t; C \mid Z_i\right)$ from the text representations of VLMs.

## 4 METHODOLOGY

### 4.1 FEATURE SPACE CONSISTENCY MAPPING

Based on the analysis of style representation disentanglement in VLMs presented in the previous chapter, our goal is to map three distinct types of information features into a shared feature space and integrate them to effectively disentangle image style features. As depicted in Figure 4(a), we use a large-scale image-text paired dataset to fine-tune the self-supervised model DINOv2. This ensures that the image content representation features generated by DINOv2 match the dimensions and scales of the representation features from the CLIP model, thus achieving feature space consistency. We select a subset of the CC3M Changpinyo et al. (2021) dataset filtered by the LLaVA Liu et al. (2024) project, containing 595,000 high-quality images and their corresponding text descriptions, as our training dataset.

Training commenced with the publicly available DINOv2 model, continuing its original training and image augmentation strategies, within a student-teacher framework. The key addition to the original model is an extra constraint: the student model's output distribution must align with the feature output distribution of the CLIP model for the corresponding text descriptions. This ensures consistency between the output spaces of DINOv2 and CLIP and enhances the model's focus on image content information.

The loss function employed during training is as follows:

$$Loss = \min_{\theta_{student}} H\left(P_{teacher}(i), P_{student}(i)\right) + \min_{\theta_{student}} H\left(CLIP_{text}(t), P_{student}(i)\right), \quad (6)$$

where where $H$ denotes the Cross Entropy Loss, $i$ represents the real image data, and $t$ stands for the paired text data. It's worth noting that the text and image representations in CLIP are already within the same feature space, thus requiring no additional alignment operations.

### 4.2 FEATURE EXTRACTION AND DECOUPLING

Subsequently, we obtained three features of the artworks through three different methods, as shown in Figure 4(b): first, image features extracted using the CLIP model which incorporates all information $I(Z_i; Z_t)$; second, image features extracted using the DINOv2 model which captures single-modal information $I\left(Z_i; C \mid Z_t\right)$ from images ; and third, text description features derived after generating image descriptions with LLMs such as GPT-4, and then extracting text information $I\left(Z_i; Z_t; C\right) + I\left(Z_t; C \mid Z_i\right)$ by using the CLIP text encoder.

We denote the image features extracted by the CLIP model as $\vec{a}$, the text features extracted by the CLIP model as $\vec{b}$, the image features extracted by the DINOv2 model as $\vec{c}$, and the final decoupled style features as $\vec{s}$. The mathematical representation of the decoupling process is as follows:

$$\vec{d} = Norm(\vec{b} + \vec{c}); \vec{s} = Norm(\vec{a} - \text{Proj}_{\vec{d}}\vec{a}) = Norm(\vec{a} - \frac{\vec{a} \cdot \vec{d}}{\vec{d} \cdot \vec{d}} \cdot \vec{d}). \quad (7)$$

As shown in Figure 4(c), the text representation generated by the CLIP model (denoted as $\vec{b}$) integrates both the content information between the image and text and unique text information. The

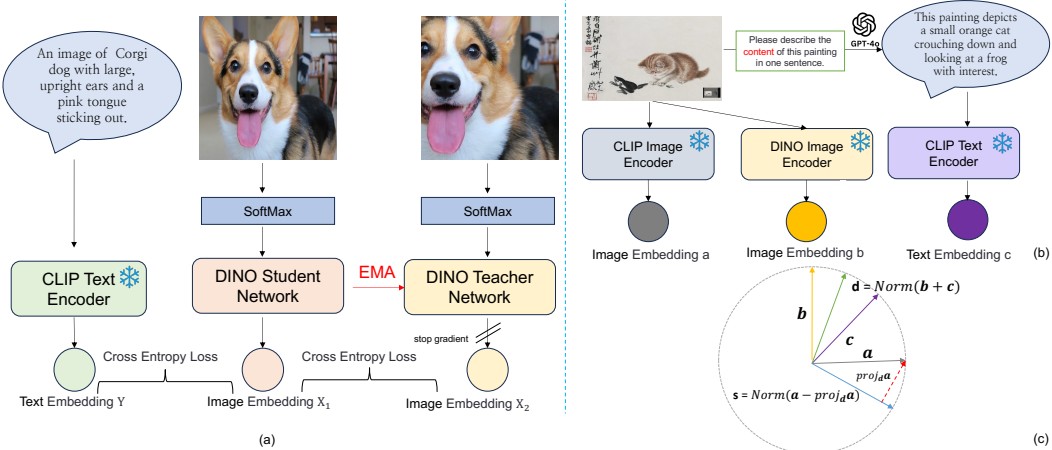

Figure 4: The feature extraction process is outlined as follows: (a) Fine-tune the DINOv2 model utilizing a large image-text dataset. The goal is to harmonize the dimensions and scales of the DINO features with those of the CLIP features. (b) Implement three unique feature extraction methodologies: CLIP image features, DINOv2 single-modal features, and CLIP text features. (c) Differentiate between image-specific information and shared information by employing orthogonal projection and subtraction operations.

DINOv2 model's representation (denoted as $\vec{c}$) focuses on capturing the image's content features. After retraining for alignment, DINOv2's representation aligns with CLIP's text representation in the same space. Thus, we can add these two representations and normalize them to obtain a composite vector $\vec{d}$, representing the image's content information.

To separate the content representation from the image representation generated by the CLIP model (denoted as $\vec{a}$) while retaining the style representation, we use the orthogonal projection technique. We first calculate the projection of $\vec{a}$ in the direction of $\vec{d}$, extracting all content features related to $\vec{d}$. Then, we subtract this projection from $\vec{a}$, leaving the component orthogonal, which represents the image's style features. Finally, we normalize this orthogonal component to ensure a consistent scale, obtaining a pure image style representation $\vec{s}$.

## 5 WeART: Precise and High-Quality Dataset for Artistic Styles

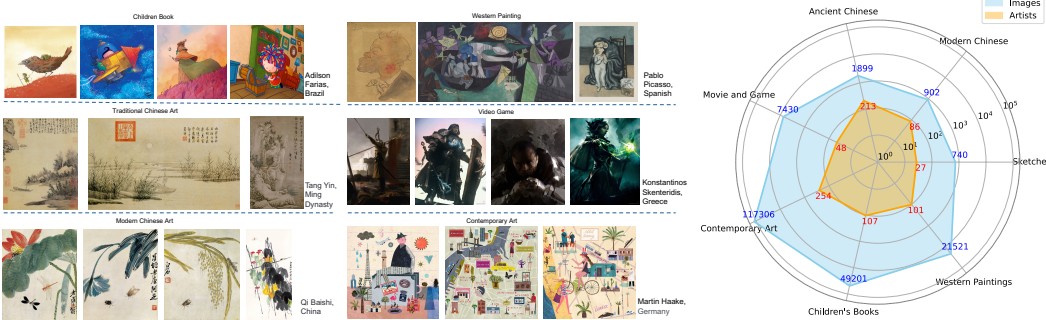

Figure 5: Artworks Displayed in the WeART Dataset and Distribution of the Number of Artists and Artworks (Log Scale).

To effectively evaluate artists' styles, it is essential to have a comprehensive dataset that spans various eras, regions, and art forms, complete with accurate author labels and high aesthetic evaluations. The largest publicly available manually annotated dataset, WikiArt, includes approximately 80,000

works from 195 artists, primarily from Europe and America, focusing on Western classical and contemporary styles. However, the limited diversity of WikiArt restricts its utility in thoroughly assessing models' capabilities.

In contrast, the LAION-Aesthetics dataset Somepalli et al. (2024), collected from the internet, is vast in scale but suffers from low-quality labels. Issues such as poor annotation quality, incorrect or missing artist labels, and imbalanced distributions are prevalent. Additionally, these datasets often feature low-resolution images with background noise, typically derived from photos or screenshots, rendering them unsuitable for precise art style evaluation tasks.

To address these issues, we constructed WeART, a high-quality dataset associating artists with their works, as shown in Figure 5. WeART contains 197,632 images from public museums, artists' blogs, and collections, covering seven major art genres and works from 661 renowned artists. Rigorous screening ensures quality and accuracy, with each artist having at least two works and over 88% having more than five. We manually removed duplicates and overly similar images, ensuring no artist has more than 2000 works. For large or irregularly proportioned artworks, such as Chinese paintings, we performed meticulous manual cropping to maintain artistic expressiveness and aesthetic quality.

WeART complements WikiArt with an artist repetition rate of less than 3%. The majority of the artworks within WeART are high-definition scans with minimal background noise, ensuring clarity and completeness. Professional curation guarantees the uniqueness and consistency of the artists' styles. WeART addresses gaps in existing art style evaluation datasets by introducing data on children's picture books and Chinese paintings, all with clear attribution to their authors. At approximately three times the size of WikiArt, WeART significantly enhances both the quality and diversity of the dataset, providing a valuable resource for the study of art styles.

# 6 EXPERIMENTS RESULTS

## 6.1 TRAINING SETTINGS

**Training Dataset:** CC3M dataset Changpinyo et al. (2021) is a large-scale collection of images paired with captions, curated to facilitate research in image captioning and visual-semantic alignment. It contains over 3 million images sourced from the web, each accompanied by machine-generated captions aimed at accurately describing the visual content.

**Settings:** The retraining experiment of the DINOv2 model on the CC3M dataset uses 4 V100 GPUs, with a batch size of 128 and 100 training epochs. The learning rate is set to 1e-5, with linear warm-up during the initial epochs followed by a cosine learning rate decay. During training, all input images are resized to a resolution of 224×224 and then augmented through normalization. All baseline models are sourced from the official open-source code repository.

## 6.2 RESULTS

To evaluate the efficacy of our proposed Decoupled Style CLIP model (DS-CLIP), we conduct a rigorous comparative analysis against the original CLIP model and a suite of VLMs. As delineated in Table 1, our DS-CLIP model, without the need for additional fine-tuning on style-specific data, demonstrates enhancements in both accuracy and recall over the original CLIP model. Moreover, it consistently excels across all benchmark assessments. Furthermore, by applying a similar decoupling strategy to the SigLIP model, we successfully augment its style retrieval proficiency. Collectively, these experimental findings underscore the universal and potent impact of decoupling content information on enhancing the stylistic representation within VLMs, without the necessity for style-labeled datasets.

In a further exploration of the decoupling technique's efficacy, we fine-tune CLIP models using the WeART dataset and then implement our decoupling approach. The comparative analysis of accuracy and recall metrics pre- and post-decoupling is presented in Table 2. Our decoupling method proves effective in bolstering the artistic style representation, even post fine-tuning. Interestingly, while models fine-tuned on larger, more generic datasets like LAION show improvement, the smaller,

| WikiArt: Query 15501/ Values: 63748 | | | | | |
|---|---|---|---|---|---|
| Model | mAP@1 | mAP@10 | mAP@100 | R@10 | R@100 |
| CLIP RN50 Radford et al. (2021) | 39.9 | 23.9 | 10.2 | 58.1 | 79.5 |
| MoCov3 ViT-B Chen* et al. (2021) | 45.8 | 27.8 | 13.0 | 70.1 | 88.3 |
| DINOv2 ViT-B Oquab et al. (2023) | 40.2 | 23.4 | 11.1 | 63.9 | 85.3 |
| DINOv2 ViT-LOquab et al. (2023) | 40.0 | 23.1 | 10.9 | 64.5 | 85.2 |
| CLIP ViT-B Radford et al. (2021) | 47.2 | 30.8 | 16.1 | 73.7 | 91.3 |
| CLIP ViT-L Radford et al. (2021) | 58.4 | 46.9 | 22.8 | 81.6 | 94.4 |
| SigLIP Zhai et al. (2023) | 58.5 | 47.2 | 22.6 | 82.1 | 94.6 |
| **DS-CLIP ViT-B** | 48.6 | 31.5 | 16.9 | 75.4 | 92.6 |
| **DS-CLIP ViT-L** | **60.6** | 48.2 | 23.4 | 82.7 | **95.0** |
| DS-CLIP SigLIP | 59.7 | **48.3** | **23.6** | **82.9** | 94.6 |
| WeArt: Query 20235/ Values 177338 | | | | | |
| CLIP RN50 Radford et al. (2021) | 39.9 | 24.8 | 18.0 | 65.7 | 85.4 |
| MoCov3 ViT-B Chen* et al. (2021) | 42.7 | 30.5 | 17.4 | 68.6 | 89.2 |
| DINOv2 ViT-B Oquab et al. (2023) | 37.3 | 32.5 | 16.6 | 77.6 | 91.1 |
| DINOv2 ViT-L Oquab et al. (2023) | 36.4 | 32.1 | 16.4 | 77.0 | 91.0 |
| CLIP ViT-B Radford et al. (2021) | 62.4 | 48.8 | 26.4 | 83.6 | 94.5 |
| CLIP ViT-L Radford et al. (2021) | 66.8 | 54.4 | 30.4 | 88.4 | 96.5 |
| SigLIPZhai et al. (2023) | 66.4 | 55.3 | 30.4 | 89.2 | 97.8 |
| **DS-CLIP ViT-B** | 63.6 | 50.0 | 27.8 | 84.8 | 94.9 |
| **DS-CLIP ViT-L** | **67.3** | **55.3** | 31.4 | **89.6** | 97.6 |
| **DS-CLIP SigLIP** | 66.9 | 55.2 | **31.6** | 89.4 | **98.1** |

Table 1: The performance of VLMs and our decoupled models on WikiART and WeART datasets.

| WikiArt | | | | | |
|---|---|---|---|---|---|
| Model | mAP@1 | mAP@10 | mAP@100 | R@10 | R@100 |
| CLIP ViT-B (FT) Radford et al. (2021) | 55.0 | 46.4 | 28.6 | 80.0 | 93.0 |
| CLIP ViT-L (FT) Radford et al. (2021) | 63.7 | 53.9 | 35.5 | 84.7 | 95.2 |
| GDA ViT-B Wang et al. (2023) | 42.6 | 32.2 | 18.1 | 67.3 | 87.1 |
| CSD ViT-B Somepalli et al. (2024) | 56.2 | 46.1 | 28.1 | 80.3 | 93.6 |
| CSD ViT-L Somepalli et al. (2024) | 64.1 | 53.5 | 35.4 | **85.7** | 95.2 |
| **DS-CLIP ViT-B** | 58.9 | 47.0 | 29.3 | 81.1 | 93.6 |
| **DS-CLIP ViT-L** | **65.2** | **54.6** | **36.2** | 85.3 | **95.4** |

Table 2: The performance of WeART fine-tuned CLIP and DS-CLIP is compared with other fine-tuned CLIP models on the WikiART dataset. The query settings are consistent with CSD.

curated WeART dataset stands out for its exceptional capacity to refine the CLIP model's stylistic representation.

## 6.3 CASE STUDY

In Figure 6, we present a comparison of image retrieval results using diverse artworks as queries, conducted with our DS-CLIP model and the standard CLIP model. In each set of displayed retrieval results, the top two items are shown side by side, with the DS-CLIP retrieval results highlighted with a blue background and the CLIP model results indicated with a yellow background. Our observations suggest that the CLIP model tends to place a strong emphasis on the content and structural features of the images during retrieval, often at the expense of considering the artistic style. This results in retrieval outcomes that may be visually similar in terms of content but differ notably in artistic style.

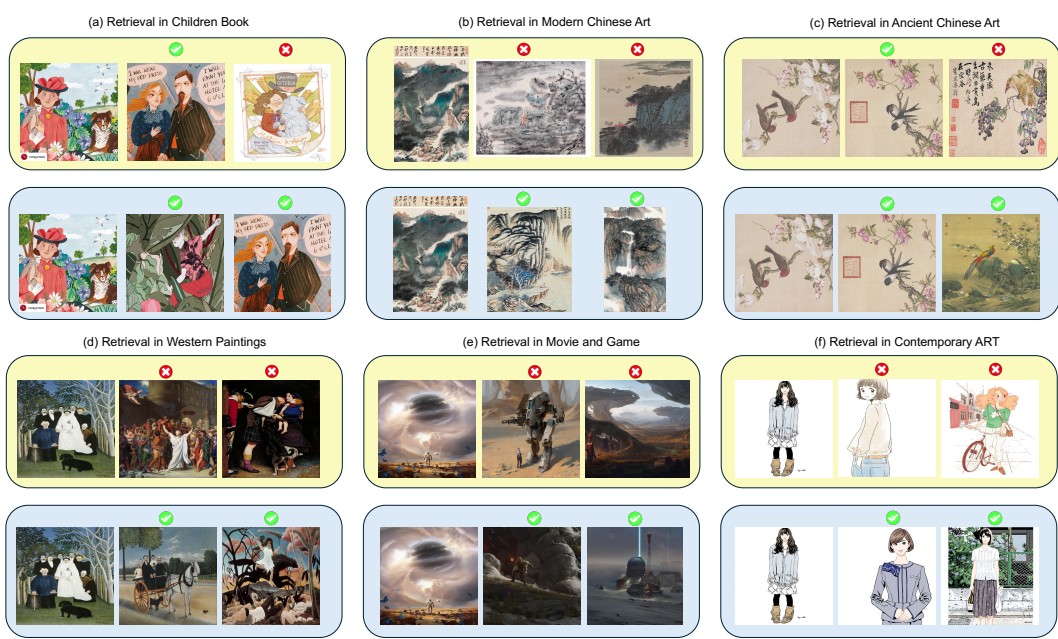

Figure 6: A case study is performed comparing the decoupled DS-CLIP/ViT-L model (blue background) with the baseline CLIP/ViT-L model (yellow background).

In contrast, the DS-CLIP model decouples artistic style features, thereby reducing the influence of content similarity on the retrieval process and enhancing retrieval accuracy.

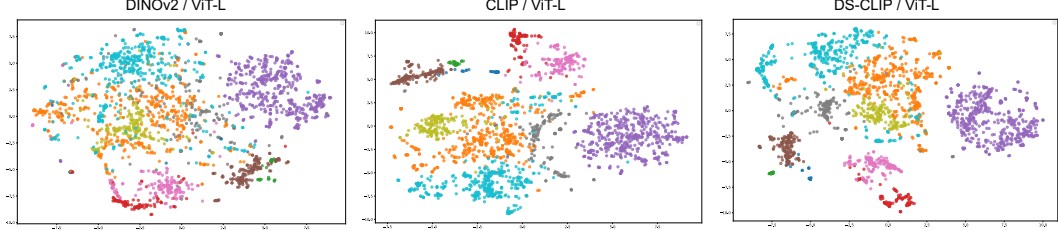

Figure 7: The t-SNE visualization is performed on the ViT-L representation models generated by DINOv2, CLIP, and DS-CLIP. Colored points represent works from 10 selected artists.

Furthermore, we present an analysis of the feature distributions of the DINOv2, CLIP, and DS-CLIP models. Using t-SNE technology Van der Maaten & Hinton (2008), in Figure 7, we visualize the ViT-L representations obtained from these three models. In the visualization results, the colored dots represent randomly sampled artworks from ten artists, with each color corresponding to one artist. The analysis reveals that, compared to the baseline model, DS-CLIP exhibits superior performance in clustering artistic styles, indicating that our model has a notable advantage in capturing and distinguishing different artistic styles.

## 6.4 MEASURING ARTISTIC STYLE IN GENERATIVE DIFFUSION MODELS

We analyze the performance of six advanced text-to-image generative models ?Podell et al. (2023); Chen et al. (2023); Sauer et al. (2024) in replicating artistic styles. Using one thousand works by two hundred artists from WikiART and WeART, we generate three variants per work with each model, resulting in a dataset of 21,000 images. The prompt "Draw [specific content description] in the style of [artist's name]" guides the generation process, and the DS-CLIP score evaluates the results.

Figure 8 illustrates that these models encounter challenges with Chinese paintings and sketches, especially when replicating the styles of ancient Chinese masters. Conversely, they demonstrate proficiency in capturing the styles of Western artists such as Van Gogh and Picasso, as well as those of contemporary illustrators. Among the evaluated models, Flux.1 stands out for its artist style representation capabilities, whereas Stable Diffusion 1.5 shows less effectiveness in comparison.

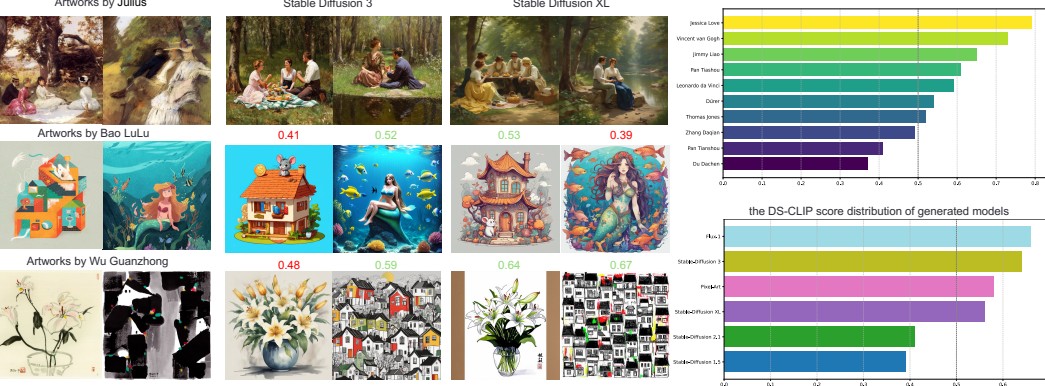

Figure 8: The case evaluation results of the generated content are presented, along with the DS-CLIP score distribution results of some artists and the score distribution results of the generative models.

## 6.5 ABLATION STUDY

In Table 3, we conduct extensive ablation studies to evaluate the effectiveness of different feature decoupling methods and their combinations. The results indicate that combining the image features decoupled from the CLIP model, which contains all information, with the text description features extracted by the CLIP model significantly improves retrieval accuracy and recall. In contrast, decoupling the image features from the untrained DINOv2 model alone leads to a substantial decrease in retrieval accuracy and recall. However, when we decouple both the trained DINOv2 image features and the CLIP text features simultaneously, the retrieval performance is optimal. All experiments are conducted on the same query dataset, query and values are the same as Table 1, with each experiment repeated three times, and the results averaged to ensure reliability.

| Ablation | | | | WikiART | | | WeART | | |
|---|---|---|---|---|---|---|---|---|---|
| $CLIP_{img}$ | $CLIP_{txt}$ | $DINO$ | $DINO_{train}$ | mAP@1 | mAP@10 | R@1 | MAP@1 | mAP@10 | R@10 |
| ✓ | | | | 58.4 | 46.9 | 81.6 | 66.8 | 54.4 | 88.4 |
| ✓ | ✓ | | | 59.7 | 47.5 | 82.2 | **67.5** | 55.0 | 89.1 |
| ✓ | | | ✓ | 59.8 | 47.3 | 81.9 | 67.2 | 54.6 | 88.9 |
| ✓ | ✓ | ✓ | | 55.3 | 43.7 | 79.5 | 61.4 | 49.9 | 84.8 |
| ✓ | ✓ | | ✓ | **60.6** | **48.2** | **82.7** | 67.4 | **55.3** | **89.6** |

Table 3: Ablation Study on the Impact of Decoupling Different Features.

## 7 CONCLUSION

In this paper, we introduce a universal decoupling method for enhancing artistic style representation. By removing text-side representations and content-related information from single-modal components within VLMs, we achieve the decoupling of artistic style. Additionally, we present the WeART dataset, a large-scale collection of artworks with high-quality annotations, designed to evaluate the artistic style representation capabilities of models. Experimental results demonstrate that our decoupling approach improves performance in style retrieval tasks across multiple datasets. For future work, we plan to incorporate additional artistic forms into our evaluation dataset and to explore more effective methods of combining representations.

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
