# OpenReview forum: "Comprehensive Artistic Style Representation for Quantitative Evaluation"
_ICLR.cc/2025/Conference — ICLR 2025 Conference Withdrawn Submission_

### Official Review · Reviewer_fBrf · 2024-10-25

**Soundness:** 2
**Presentation:** 3
**Contribution:** 3
**Rating:** 3
**Confidence:** 3

**Summary:**

This paper addresses the challenge of representing artistic style in computer vision. The main contributions are:

1) An analysis of artistic style representation from an information theory perspective, identifying limitations in current models.

2) A plug-and-play method to efficiently separate content information from style in Vision-Language Models (VLMs).

3) Introduction of the WeART dataset - a large-scale, high-quality annotated art dataset for evaluating artistic style representation.

In summary, the authors propose an approach to disentangle content from style information in VLMs, aiming to overcome the limitations of current models in generalizing across diverse artistic styles and precisely identifying individual artists' styles. The proposed method leverages differences between unimodal and multimodal models in representing artistic style. The paper combines theoretical analysis with practical implementation and evaluation, while also providing a new dataset to support future research in this area.

**Strengths:**

1. The paper presents a new approach to disentangle content from style information in Vision-Language Models (VLMs).

2. The authors propose a plug-and-play method for separating content information, which suggests an innovative and flexible solution to a complex problem.

3. The paper leverages information theory to analyze artistic style representation.

4. The introduction of the WeART dataset could provide a new benchmark for the field.

5. The paper's structure is well-organized, and clearly presents problem statement, theoretical analysis, methodology, and experimental results.

6. The Figures 1 and 2 help to illustrate concepts and results effectively.

**Weaknesses:**

1. Limited comparison with existing methods:

The comparisons with state-of-the-art methods in artistic style representation are not sufficient. To strengthen this aspect, the authors are suggested to:

(1) Conduct a comparison with methods including ArtFID [A], the attribution model by Wang et al. [B], and the style descriptor framework by Somepalli et al. [C].

[A] Matthias Wright and Bjo ̈rn Ommer. Artfid: Quantitative evaluation of neural style transfer. DAGM German Conference on Pattern Recognition 2022.

[B] Wang et al. Evaluating data attribution for text-to-image models. ICCV 2023.

[C] Somepalli et al. Measuring style similarity in diffusion models. Arxiv 2024.

(2) Provide quantitative results comparing the proposed method's performance against these baselines on standard datasets like WikiART, as well as the new WeART dataset.

2. Lack of ablation studies:

There're no ablation studies to validate the contribution of each component in the proposed method. The authors are suggested to:

(1) Conduct ablation experiments to isolate the impact of the decoupling technique VS other aspects of the proposed approach.

(2) Analyze how different components affect performance on various artistic styles or genres.

3. Potential bias in the WeART dataset:

Although introducing a new dataset is valuable, there's no discussion of potential biases. The authors are suggested to:

(1) Provide a detailed breakdown of the dataset composition, including artist demographics, art styles, and historical periods represented.

(2) Discuss any potential biases in the dataset selection process, and how these biases might impact model performance or generalizability.

(3) Compare the characteristics of WeART with existing datasets like WikiART to highlight its unique contributions of WeART dataset.

4. Limited exploration of failure cases:

There lacks an analysis of cases where the proposed method might underperform. The authors are suggested to:

(1) Discuss failure cases or limitations of the proposed approach.

(2) Provide examples of artistic styles or scenarios where the proposed method struggles and analyze the reasons behind these challenges.

5. Lack of human perceptual evaluation:

The paper seems to focus on quantitative metrics for style retrieval, but doesn't mention any human perceptual studies. The authors are suggested to:

(1) Conduct a user study with art experts to evaluate the quality and accuracy of the style representations.

(2) Compare the proposed method's style classifications with human judgments, to validate its alignment with human perception of artistic style.

6. Limited discussion on computational efficiency:

The paper did not mention the computational efficiency of the proposed method. The authors are suggested to compare the computational cost of the decoupling technique to the existing methods.

7. Insufficient exploration of cross-domain generalization:

Although the method is tested on multiple datasets, the paper did not explicitly mention its performance across different art forms. The authors are suggested to evaluate the proposed method's performance on diverse art forms (e.g., digital art, sculptures, architecture) to test its cross-domain generalization ability.

**Questions:**

1. Methodology Clarification:

(1) Can you provide more details on how your decoupling technique specifically separates content from style information in VLMs? A step-by-step explanation or pseudocode would be helpful.

(2) How does your method handle cases where style and content are closely intertwined, such as in abstract art?

2. Performance Metrics:

(1) What specific metrics did you use to evaluate style retrieval performance? How do these metrics capture the nuances of artistic style beyond simple classification accuracy?

(2) Did you consider any perceptual metrics or human evaluation studies to validate your results?

3. Comparison with Existing Methods:

(1) Could you provide a more detailed comparison of your method's performance against state-of-the-art approaches like ArtFID [A] or the work by Somepalli et al. [B]?

[A] Matthias Wright and Bjo ̈rn Ommer. Artfid: Quantitative evaluation of neural style transfer. DAGM German Conference on Pattern Recognition 2022.

[B] Somepalli et al. Measuring style similarity in diffusion models. Arxiv 2024.

(2) How does your method perform on standard benchmarks used in these previous works?

4. WeART Dataset:

(1) Can you provide more details on the construction process for the WeART dataset? How did you ensure diversity and representation across different artistic styles, periods, and cultures?

(2) How does the WeART dataset compare to existing datasets like WikiART in terms of size, diversity, and quality of annotations?

(3) Is there a plan to make WeART publicly available, and if so, under what terms?

5. Generalization and Robustness:

(1) How well does your method generalize to artistic styles not represented in the training data?

(2) Have you tested the robustness of your approach to variations in image quality, cropping, or other transformations that might occur in real-world scenarios?

6. Computational Efficiency:

(1) What is the computational efficiency of your method compared to the existing approaches?

(2) Is your decoupling technique suitable for real-time applications, or is it primarily designed for offline analysis?

7. Practical Applications:

(1) Apart from style retrieval, have you explored other potential applications of your method, such as style transfer or artwork authentication?

8. Limitations and Failure Cases:

(1) What do you consider to be the main limitations of your current approach?

(2) Are there specific artistic styles or types of artworks where your method struggles? If so, what do you think are the reasons for these challenges?

---

### Official Review · Reviewer_zVdy · 2024-11-02

**Soundness:** 1
**Presentation:** 1
**Contribution:** 1
**Rating:** 1
**Confidence:** 5

**Summary:**

The paper presents two main contributions.  First, a method to learn a disentangled representation of artistic style using a combination of CLIP and DINOv2.    As with prior work in this problem space, the goal is to learn a representation the separates the appearance i.e. style from the content i.e. subject matter in an image. Second, a dataset called WeART comprising ~197K artist images  from museums, blogs etc. covering ~660 artists is contributed.

**Strengths:**

Learning a robust, generalizable encode for disentangled style representation remains an open problem worth of study.

**Weaknesses:**

The paper contains major flaws in several respects.

First, the paper is not well contextualized within prior work.  A new style dataset (WeART) is proposed, but there is no review of prior style datasets beyond WikiArt (2015) to argue for its uniqueness or value.   For example, the authors state that WikiArt contains only 80K artworks (it has 81K)  and ~200 artists vs. the proposed WeART 197K artworks and 600 artistis.  But there is OminArt (432K artworks), BA M (2M artworks),  BAM-FG (2.6M artworks in fine-grained artistic variations), SemArt, AVA, ArtEmis, the list goes on.  The authors should not only cite and argue for the benefit of their new dataset against prior datasets, but also evaluate their proporsed method against them.    In addition to the lack of comprehensive study of prior datasets, the paper also does not cite or compare the proposed new style representation  learning approach to relevant baselines.  There are several techniques in common use e.g. ALADIN, ALADIN-NST, StyleBabel, MUNIT based approaches, etc. all that seek to learn disentangled representations.

Second, there is unnecessary formalism around information theory introduced within the paper (Section 2) that is then unused.  Specifically the paper makes an opening argument about the need to consider information theory in the design of a disentangled style representation.  Notation is introduced to motivate this argument, but is not actually used in the method itself nor is any information theory used in the training of the method.  Rather, Section 2 culminates in a short quantitative study which appears to show retrieval results using various CLIP (multimodal VLM) and DINO (self-supervised VLM) derivatives with no reference to the formalisms introduced in Section 2.  Overall it is very unclear where a main thrust of the paper in its information theory based motivation delivers in the method.  More generally, the definition of style itself is very vague in this paper (Section 1) relying upon quotation and phrases such as ‘essence of an artists personalized expression’.  It would be better to make a clear definition of style (appearance) vs. content (subject matter) and align the work with the body of prior work exploring disentangled content/style learning.

Third, the technical method itself seems to lack novelty and is poorly justified.  The method is essentially extracting visual features from an image using CLIP and DINOv2 and then captioning the image with ‘a VLM such as GPT-4’ (it is never stated which LLM is actually used in the experiments)  and using CLIP to encode the captioned text.  It is not justified why the LLM would produce a style caption that leads to learning of a disentangled representation.  Rather there is a reliance on the ‘orthogonal projection technique’ which seems to imply that vector math in the CLIP/DINO hybrid space (which appears to be some concatenated, normalized vector of the two) can somehow disentangle style and content – this seems a circular argument since developing a space with such properties is infact the goal of the learning.
Also, as a matter of clarity - The paper introduces the name ‘DS-CLIP’ (Decoupled Style CLIP) in Section 6 as the name of the proposed method, which would have been much clearer to state in the introduction in terms of readability and explaining the core motivation of the paper at the outset.  The discussion of large VLMs throughout the methodology seems to obfuscate this end goal, whereas the LLM aspect of the work is simply using LLM (such as GPT) to caption the images.

Fourth, the paper targets ‘comprehensive’ style, implying that prior style representation learning work is not able to generalize to a comprehensive gamut of styles.  However the paper does not evaluate or study most available style datasets, and does not baseline against common prior style representation learning techniques.  It is unclear how well the paper therefore delivers on this core claim of improved generality.

**Questions:**

Overall the paper contains serious flaws in its technical method, and in the clarity of the exposition which at times serves to obfuscate the core contribution.  There are clear deficiencies in the treatment of prior work in style learning, both in terms of prior datasets and prior methods.  For me, the paper is a clear rejection case.

---

### Official Review · Reviewer_tBNf · 2024-11-03

**Soundness:** 2
**Presentation:** 2
**Contribution:** 2
**Rating:** 3
**Confidence:** 4

**Summary:**

This paper analyzed the content and style disentanglement from the information theory perspective. The authors proposed a decoupling method for disentangling style and content in Vision-Language Models (VLMs) based on information theory, which improves the performance of style retrieval tasks. This paper also introduces a new dataset, WeART, a large-scale art dataset with high-quality annotations for evaluating artistic representations of VLMs.

**Strengths:**

1. This paper introduces a new large-scale high-quality dataset, WeART. The dataset improves the diversity and quality over current datasets.
2. The paper presents a novel method for disentangling style and content in Vision-Language Models (VLMs) based on information theory.
3. This paper examines the problem of content and style disentanglement from an information theory perspective, offering a novel approach to this topic.

**Weaknesses:**

1. Although the authors discuss style transfer in the related work section, they do not experiment with style transfer tasks, which are crucial for the proposed method's aim of decoupling content from style representations.
2. The WeART dataset comprises only seven genres, which may not provide compelling evidence for the representation of fine-grained styles frequently employed in practical applications.
3. The analysis of mutual information between style and content in multimodal data lacks sufficient theoretical support or deductive proof. Specifically, Equations 1 and 2 appear to be based on the assumption that image information can be directly divided into content and style components, yet this division is not adequately substantiated.
4. There are some formatting issues and problems with image references that slightly hinder the reading experience.

**Questions:**

1. Regarding Equation 1, why would the style information conveyed by an image differ from that conveyed by its corresponding text description? If the text accurately describes the image's style, both should theoretically carry the same stylistic information.
2. In the experimental section, the baseline model appears to retrieve the correct style as well, such as in example (c), where the style of the right image more closely matches the style of the query image than the proposed model does.
3. In Section 6.4, how is the DS-CLIP score calculated?
4. Artistic style resemblance is inherently subjective, particularly in user-oriented tasks such as style transfer and style generation. It would be more persuasive for the authors to consider including a human evaluation to compare their proposed evaluation metrics with human judgments.

**Details Of Ethics Concerns:**

No ethics concerns.

---

### Official Review · Reviewer_7fsc · 2024-11-03

**Soundness:** 3
**Presentation:** 1
**Contribution:** 2
**Rating:** 3
**Confidence:** 4

**Summary:**

This article presents a novel method for extracting artistic style features from images. It aims to resolve the issues of inadequate and redundant feature extraction found in earlier methods. The author processed the feature vectors obtained from a large model after fine-tuning to isolate the "content information" within the features, allowing for the extraction of pure artistic style attributes.

This paper should be rejected for several reasons: (1) the theory proposed by the authors is not adequately supported by experimental evidence; (2) the experiments lack clarity and motivation, and do not provide significant contributions; (3) the overall presentation of the paper is imprecise and unrefined; and (4) some of the proposed methods are not relevant to the problems addressed in the paper.

**Strengths:**

The feature extraction method introduced in this article is innovative, and its effectiveness surpasses that of other large models.
Additionally, the author created a large-scale, high-definition dataset of art style images, which significantly contributes to the field of style transfer.

**Weaknesses:**

The author did not provide rigorous proof for the hypothesis he proposed, and subsequent experiments were unable to validate his theory. While the author believes that the adjustments he made can enhance the model's ability to extract style features from images, the experimental results suggest that the model may only be learning a single correspondence, such as image-text matching, rather than genuinely identifying the style features.

The experimental results do not support the formula proposed by the author, and the author has not provided evidence for their theory, which makes it unconvincing. If the author's hypothesis is valid, more intuitive experiments, such as style transfer, should be conducted to demonstrate this. Alternatively, a logical explanation for the proposed formula should be provided.

**Questions:**

1. The author did not clarify how to obtain the text corresponding to the images in the dataset. Relevant descriptions should be provided if the data is sourced through a large model.
2. The author did not explain the rationale behind DINOv2's fine-tuning process. Would CLIP not be more straightforward to align the feature sizes?
3. Several aspects of the fine-tuning process for DINOv2, such as the use of the teacher-student network structure and the design of the loss function, are not explained.
4. Why was DINOv2 chosen for fine-tuning and subsequent experiments? The scatter plot in Chapter 6 indicates that CLIP performs well, whereas DINOv2 shows poor performance. The author's fine-tuning of DINOv2 did not significantly improve the model.
5. If the hypothesis presented in the article is valid, the author should conduct visual experiments, like style transfer, to validate their theory. The experimental results based solely on the article's combination of text and images are not very convincing.
6. Figure 2 introduces the underlying assumptions and analysis, while Part 3 explains these assumptions rather than providing proof. More specific evidence or references would be beneficial as supplementary material.
7. Section 4.1 discusses aligning DINOv2 and CLIP text encoders in the feature space to enhance the model's attention to content information. Does this imply that CLIP text encoding represents content information? Why hasn’t the shared style information been given attention?
8. The article does not state why GPT-4o was used to generate text descriptions.
9. This question is similar to question 7 regarding formula 7, yet the article does not clarify why “d” is considered content information.

---

### Note · Authors · 2024-11-26

I have read and agree with the venue's withdrawal policy on behalf of myself and my co-authors.